# A Comparison of ^68^Ga-PSMA PET/CT-Based Split Renal Function with ^99m^Tc-MAG3 Renography in Patients with Metastatic Castration-Resistant Prostate Carcinoma Treated with ^177^Lu-PSMA

**DOI:** 10.3390/diagnostics14060578

**Published:** 2024-03-08

**Authors:** Lerato Gabela, Thokozani Mkhize, Bawinile Hadebe, Lerwine Harry, Thembelihle Nxasana, Nontobeko Ndlovu, Venesen Pillay, Sphelele Masikane, Maryam Patel, Nozipho Elizabeth Nyakale, Mariza Vorster

**Affiliations:** 1Department of Nuclear Medicine, College of Health Sciences, University of KwaZulu Natal, Private Bag X54001, Durban 4001, South Africa; thokozani.mkhize@ialch.co.za (T.M.); hadebeb1@ukzn.ac.za (B.H.); lharry27@gmail.com (L.H.); lihleskhonza@gmail.com (T.N.); venesen.pillay@gmail.com (V.P.); sphetiba@gmail.com (S.M.); maryam.patel@ialch.co.za (M.P.); vorsterm1@ukzn.ac.za (M.V.); 2Inkosi Albert Luthuli Central Hospital, Durban 4001, South Africa; 3Department of Nuclear Medicine, Sefako Makgatho Health Science University, Pretoria 0208, South Africa; n.nyakale@yahoo.com

**Keywords:** PET/CT, ^68^Ga PSMA-11, ^99m^Tc-MAG3, kidney, renal function

## Abstract

Background: Physiological PSMA expression in the cells of the proximal renal tubules and consecutive radiopharmaceutical binding and retention could potentially lead to radioligand-therapy-induced nephrotoxicity. Thus, patients with metastatic castration-resistant prostate cancer undergo ^99m^Tc-Mercaptoacetyltriglycine (MAG3) renal scintigraphy to assess kidney function and to exclude renal obstruction as part of their workup for PSMA-targeted radioligand therapy (RLT). ^99m^Tc-MAG-3 renal scintigraphy often requires an additional visit to the nuclear medicine department and patients spend 30–90 min in the department, which is inconvenient and takes up camera time. In addition, the patients are subjected to a baseline ^68^Ga-PSMA PET/CT to assess for PSMA-positive disease prior to targeted radioligand therapy. The aim of this retrospective cross-sectional study was to compare ^99m^Tc-MAG-3-based split renal function (SRF) with ^68^Ga-PSMA-derived SRF. Methods: This retrospective cross-sectional study included 28 patients with histologically proven metastatic castration-resistant prostate cancer (mCRPC) who received 177Lu-PSMA617. A comparison between the split renal function using ^68^Ga-PSMA PET/CT and the ^99m^Tc-MAG-3-derived split renal function was carried out in 56 kidneys (*n* = 56). The SRF on ^68^Ga-PSMA was calculated using the volume and the average standard uptake value (SUVmean) within each VOI calculated as previously described by Roser et al.: *SRF* = (*VOLUME_right_*) ∗ *SUVmean_right_*/(*VOLUME_right_* ∗ *SUVmean_right_* + *VOLUME_left_* ∗ *SUVmean_left_*). Paired tests and correlation coefficients were used to compare ^68^Ga-PSMA and ^99m^Tc-MAG-3. A visual comparison of kidney morphology on both studies was also performed. Results: The median SRF of the right kidney was 49.9% (range: 3–91%) using ^68^Ga-PSMA PET/CT and 50.5% (range: 0–94%) with ^99m^Tc-MAG3 scintigraphy. Notably, there was a strong correlation between SRF measurements obtained from PSMA and ^99m^TcMAG3, with a Pearson correlation coefficient of 0.957 (*p* < 0.001). Both ^99m^Tc-MAG3 and ^68^Ga-PSMA PET/CT studies identified morphological renal abnormalities; there were nine hydronephrotic kidneys, four shrunken kidneys and one obstructed kidney, and there was a strong positive correlation between ^68^Ga-PSMA kidney morphology and ^99m^TcMAG3 renal scintigraphy kidney morphology, with a correlation coefficient of 0.93. Conclusions: PSMA-derived split function demonstrated a high correlation with renal function assessed on diuretic ^99m^Tc-MAG3 renograms. PET-derived split renal function may, therefore, be considered an alternative to diuretic renogram-based split function. Furthermore, both ^99m^Tc-MAG3 and ^68^Ga-PSMA PET/CT studies identified morphological renal abnormalities such as hydronephrosis, shrunken and obstructed kidneys. This correlation underscores the potential utility of ^68^Ga-PSMA imaging as a valuable tool for assessing kidney morphology as an alternative to renogram split function in clinical practice.

## 1. Introduction

Prostate cancer is one of the commonest cancers in men worldwide, and the second leading cause of cancer mortality, after lung cancer, in men globally [1]. In South Africa, 1 in 28 men are affected, with more than 20% of diagnosed patients dying from the disease. African men are six times more likely to develop prostate cancer compared to those in most developed countries, and the disease tends to follow a more aggressive course [2] due to multiple factors such as genetic alterations, protein differences, tumour microenvironment, and even circulating hormones and vitamins that might contribute to the differing phenotypes [3,4,5]. These differences in tumour biology result in more rapid prostate tumour growth and earlier transformation from indolent to aggressive disease [2]. In addition, cultural factors and poverty result in delays in accessing healthcare services; as a result, the patients present at more advanced disease stages, which may render them unsuitable for localised therapies [3]. 

Metastatic castration-resistant prostate cancer (mCRPC) is conventionally treated with chemotherapy and immunotherapy; however, the survival benefits from these agents are modest, with survival of less than 20 months [6]. 

The modern clinical management of prostate cancer relies on exploiting the prostate-specific membrane antigen (PSMA) as a molecular target for both the imaging and treatment of prostate cancer in patients. PSMA is a type II transmembrane glycoprotein that is abundantly overexpressed on the surface of prostate cancer cells within the neovasculature of other solid tumours, with limited expression in most normal tissues, e.g., proximal convoluted tubules of the kidneys, establishing the basis for the selective targeting of prostate cancer lesions [7,8].

Lutetium-177 (^177^Lu) is a beta-emitting radioisotope used in radionuclide therapy, which, when labelled with PSMA peptides, is used for the treatment of metastatic castration-resistant prostate cancer (mCRPC) [9,10,11]. ^177^Lu-PSMA has been shown to be effective in 30–70% of patients with prostate cancer with several men experiencing >50% of reduction in serum PSA [9,11,12,13]. 

The VISION trial showed evidence of the safety and efficacy of ^177^Lu-PSMA-617 in combination with standard of care in patients with metastatic castration-resistant prostate cancer whose disease had progressed after or during at least one taxane regimen and at least one novel androgen-axis drug, which led to FDA approval of ^177^LuPSMA-617 [6]. This was also shown in the TheraP trial, which compared ^177^Lu-PSMA-617 against best supportive/best standard of care (Cabazitaxel) in men with mCRPC; ^177^Lu-PSMA-617 led to a higher prostate-specific antigen (PSA) response (66% vs. 44%) and fewer grade 3 or 4 adverse events [14]. 

In a multicentre study performed in Germany by Rahbar et al., adverse events associated with ^177^Lu-PSMA-617 comprised grade 3–4 hematologic adverse events in 18 of 145 patients (12%). One patient experienced severe leukopenia, eleven (8%) patients anaemia, two (2%) patients had thrombocytopenia and four patients experienced a combination of these conditions. No grade 3 or 4 nephrotoxicity occurred. Mild-to-moderate xerostomia was reported for 11 (8%) patients overall [7]. 

The radiation dose to kidneys has been reported to range from approximately 0.4 to 1.0 Gy per GBq administered, with figures in the range of 0.6 Gy/GBq being the most consistent [15,16]. In a study by Ngoc et al., only a slight reduction in renal function was seen in 50% of patients undergoing RLT with ^177^Lu-PSMA-617 and correlated with cumulative doses [17]. Acute kidney injury and significant renal function loss (grade 3) was observed in patients with sub-renal obstruction, which highlights the importance of performing imaging prior to RLT to identify patients at risk of nephrotoxicity. In a metanalysis by Sadaghiani et al., the pooled estimated proportion of patients with grade 3 or 4 adverse events for all the evaluated toxicities including nephrotoxicity was <10% [18]. 

In addition to the ^68^Ga-PSMA PET/CT routinely carried out to assess tumour PSMA expression ^177^Lu-PSMA prior to radioligand therapy, ^99m^Tc-Mercaptoacetyltriglycine (MAG3) renal scintigraphy is also performed to assess baseline kidney function and to exclude renal obstruction [2]. 

Numerous methods allowing for the measurement of individual kidney function have been described and commonly used investigations for renal function include measurement of glomerular filtration rate (GFR) by ^99m^Tc-DTPA and measurement of the tubular extraction rate using ^99m^Tc MAG3 scintigraphy, as well as serum urea and creatinine.

Estimating the glomerular filtration rate (eGFR) is the simplest method to assess renal function in clinical practice using the modification of diet in renal disease (MDRD) formula based on the renal retention parameter creatinine [19]. However, this method does not provide information on the relative distribution of renal function in each kidney and drainage, and this is vital in patients undergoing targeted radionuclide therapies. 

^99m^Tc-MAG-3 renal scintigraphy is well established for the assessment of renal function. It is a tracer with a high renal extraction [20,21] that is highly protein-bound and is removed from the plasma primarily by organic anion transporter 1 (located on the basolateral membrane of the proximal renal tubules). The extraction fraction of ^99m^TcMAG3 is 40–50%, which is more than twice that of ^99m^Tc-DTPA. In light of its more efficient extraction, ^99m^Tc-MAG3 is preferred over ^99m^Tc-DTPA in patients with suspected obstruction and impaired renal function [22]. It is a non-invasive, dynamic, and widely available test which is used to differentiate between dilated non-obstructed and dilated obstructed upper urinary tracts [23]. Renography is clinically indicated in the measurement of differential renal function (DRF), also known as split renal function, which describes the relative function of each kidney. This is indicated in patients with signs or symptoms of obstruction and to rule out renal obstruction in asymptomatic patients with signs of hydronephrosis detected radiologically. It provides information on excretion and renal function, which guide therapeutic decisions [23]. MAG3 renography is based on renal function and tracer washout from the collecting system with urine flow stimulated by the administration of furosemide [22]. Patients are required to undergo ^99m^Tc-MAG-3 renal scintigraphy prior to ^177^Lu-PSMA radioligand therapy and during follow-up. They often require additional visits to the nuclear medicine department, which is inconvenient and time consuming, and leads to additional radiation doses and camera time. 

PSMA is physiologically overexpressed in the proximal tubular cells of the kidneys and ^68^Ga-PSMA is excreted via the renal system [24]. Several studies have shown the utility of PSMA in the imaging of renal function [25,26,27,28]; therefore, we hypothesised that ^68^Ga-PSMA PET/CT imaging could be used as a surrogate for the determination of PSMA-PETderived split renal function (SRF) at baseline, to assess kidney function and detect changes during follow-up, and to rule out obstruction induced by tumour or renal dysfunction caused by PSMA-targeted radioligand treatment. 

We subsequently compared ^99m^Tc-MAG-3-based SRF with ^68^Ga-PSMA-PET-derived SRF in patients with mCRPC who had undergone both imaging modalities. 

## 2. Materials and Methods

This is a retrospective cross-sectional study involving 28 patients with metastatic castration-resistant prostate cancer (mCRPC) who previously received first- and second-line chemotherapy in addition to androgen deprivation therapy, who underwent ^177^LuPSMA-617 radioligand therapy between June 2019 and September 2023. A comparison between the split renal function using ^68^Ga-PSMA PET/CT and ^99m^Tc-MAG-3-derived split renal function was carried out in 56 kidneys (*n* = 56). ^68^Ga-PSMA PET/CT and ^99m^Tc-MAG3 scintigraphy were performed within a median of 4 weeks, with a range of 1–6 weeks. The inclusion criteria included all adult males with castration-resistant prostate cancer eligible for ^177^Lu-PSMA therapy with PSMA expression of most lesions as determined by PSMA-targeted imaging, availability of a ^99m^Tc-MAG 3 renogram, eGFR, urea and creatinine. 

This study was approved by the Human Research Ethics Committee of the University of KwaZulu Natal (protocol reference number: BREC/00003636/2021). All procedures were performed in accordance with the ethical standards of the institutional research committee in alignment with the 1964 Declaration of Helsinki and its latter amendment. 

### 2.1. Imaging Procedure

#### 2.1.1. ^68^Ga-PSMA PET/CT

All patients underwent a ^68^Ga-PSMA 11-PET/CT scan, which included a 5 mm thick slice standard for low-dose unenhanced CT scan for attenuation correction and anatomical localisation. Each patient received an intravenous activity with a median of 110.9 MBq ranging from 75.5 to 179.4 MBq and the time between injection and PET image acquisition ranged from 55 to 65 min. No contrast agents or diuretics were administered. 

The PET data were acquired using a Biograph mCT S(64) PET/CT scanner (Siemens Healthineers, Erlangen, Germany) with a Flow Motion Bed Speed of 0.15 mm/s (5 min per Bed), covering a wide 21.4 cm field of view (FOV). Iterative reconstruction of the acquired images was conducted using the Siemens Flash 3D algorithm specifically employing a 3D OSEM algorithm with 3 iterations, 24 subsets and a Gaussian filter of 5 mm. 

#### 2.1.2. ^68^Ga-PSMA PET/CT Image Analysis

To determine the relative distribution of ^68^Ga-PSMA in both kidneys, volumes of interest (VOIs) were manually delineated using the syngo.via workstation (Siemens Healthineers).

Subsequently, the volume and the mean standard uptake value (*SUVmean*) within each VOI were used to calculate the split renal function (*SRF*) as previously described by Roser et al. [29]: *SRF* = (*VOLUME_right_*) ∗ *SUVmean_right_*/(*VOLUME_right_* ∗ *SUVmean_right_* + *VOLUME_left_* ∗ *SUVmean_left_*)


#### 2.1.3. ^99m^Tc-MAG3 Renal Scintigraphy

All patients (*n* = 28) underwent renal scintigraphy imaging with 185 MBq of ^99m^Tc-MAG3 injected intravenously. Dynamic images were acquired at the time of injection. These were followed by a 1 min static image immediately after the dynamic (premicturition) and after asking the patient to void (post micturition) using the Siemens EVO gamma camera. Manual regions of interest (ROI) were drawn around each kidney with background correction applied. The SRF(_MAG3_) was calculated in the first 2 min of the study (second frame) using the integral method and expressed as a percentage. Furosemide was administered at 15 min post tracer injection (F + 15) in cases of renal retention. 

### 2.2. Statistical Analysis

The statistical data analysis was conducted in R Statistical computing software of R Core Team, 2020, version 3.6.3. Depending on the distribution of the data, the descriptive statistics of numerical measurements were summarised as ranges, minimum, maximum, quartiles, median, means, standard deviation and the coefficient of variation. In addition, boxplots and histograms were used for the visual display of the descriptive patterns. Paired Student’s *t*-tests and Pearson correlation coefficients were used to compare the ^68^Ga-PSMA and ^99m^Tc-MAG-3.

All the inferential statistical analysis tests were conducted at a 5% level of significance. 

## 3. Results

### 3.1. Patient Characteristics

The patients’ characteristics are summarised in Table 1. A total of 28 patients with a median age of 67.5 and a range of 45–86 with mCRPCa were included. Fifteen (15%) (4/28) of the patients had chronic kidney disease (CKD) with eGFR < 60 mL/min (CKD stage 3A *n* = 3; stage 3B *n* = 1), of which one had a non-functional kidney, one had a poorly functioning and dilated kidney, one had a small and poorly functioning kidney, and one had an obstructed kidney. Eighty-five percent (85%) of patients (24/28) had normal renal function (eGFR > 60 mL/min). The Gleason scores were as follows: 6 in 3.6%, 7 in 35.7%, 8 in 14.3% patients, 9 in 28.6% patients, and 10 in 7.1% of patients. The median creatinine was 76 µmol/L with a range of 52–205, and the median PSA was 359.07 µmol/L with a range of 3.95–6685. The median urea was 5.25 mmol/L (with a range of 2.82–8.1), as shown in Table 1 and Figure 1 and Figure 2. The patients underwent ^99m^Tc-MAG3 and ^68^Ga-PSMA within two weeks of each other. All 28 patients had bone metastases; in addition, 19 patients also had metastatic lymph node involvement, and 2 patients had lung metastases. 

### 3.2. Comparison of SRF for ^99m^Tc-MAG3 and PSMA

SRF measurements were taken using ^68^Ga-PSMA PET/CT and ^99m^Tc-MAG3 scintigraphy. The median SRF of the right kidney was 49.9% (range: 3–91%) using ^68^Ga-PSMA PET/CT and 50.5% (range: 0–94%) with ^99m^Tc-MAG3 scintigraphy, as shown in Table 2 and Figure 3, Figure 4 and Figure 5. The correlation between SRF measurements obtained from ^68^Ga-PSMA and ^99m^Tc-MAG3 was highly significant (*p* < 0.001), with a correlation coefficient of r = 0.957. There was excellent agreement between the two modalities. 

### 3.3. Comparison of Morphological Abnormalities

Concordance was also noted between morphological renal abnormalities on ^99m^TcMAG3 renal scintigraphy and ^68^Ga-PSMA PET/CT studies. There was a strong positive correlation between ^68^Ga-PSMA kidney morphology and ^99m^Tc-MAG3 renal scintigraphy kidney morphology, with a correlation coefficient of 0.93, as shown in Figure 6. Both studies identified all hydronephrotic kidneys (*n* = 9) and shrunken kidneys (*n* = 4) that were poorly functioning or non-functional. Figure 7 shows a patient with normal renal function and symmetrical SRF on both modalities. 

Figure 8 shows an example of a patient with a non-functioning right kidney with hydronephrosis on ^68^Ga-PSMA which showed no uptake on ^99m^Tc-MAG3 scintigraphy. One patient exhibited kidney obstruction on ^99m^Tc-MAG3 scintigraphy, which corresponded to suspected obstruction on ^68^Ga-PSMA, as demonstrated in Figure 9. 

Additionally, three patients had kidney cysts detected visually on ^68^Ga-PSMA. 

## 4. Discussion

In patients undergoing RLT, there is concern about potential nephrotoxicity secondary to PSMA expression on the renal tubules, which results in radiopharmaceutical retention [15,29], and, therefore, renal scintigraphy with ^99m^Tc-MAG3 is routinely performed as a baseline study to assess kidney function and exclude kidney obstruction before therapy. ^99m^Tc-MAG3 is the agent of choice for renal scintigraphy as it provides excellent image quality even in the presence of severely decreased renal function [30]. In this retrospective study, a group of 28 prostate cancer patients with a mean age of 64 years underwent kidney evaluation using two separate methods: ^68^Ga-PSMA-11 PET/CT and standard ^99m^Tc-MAG3 kidney scintigraphy. Our results show a strong positive correlation between ^68^Ga-PSMA-derived split function and renal function assessed through imaging with ^99m^Tc-MAG3 renal scintigraphy, with a correlation coefficient of r = 0.957 (*p* < 0.001), and similarity in median SRF values was also noted. This suggests that ^68^Ga-PSMA-derived split function can serve as a reliable alternative for assessing functional renal tissue. Our findings are in agreement with previously published data which showed a positive correlation between SRF derived by ^68^GaPSMA PET and ^99m^Tc-MAG3 scintigraphy as shown in Table 3 [25,26,27,28]. In a study by Rassek et al., SRF_MAG3_ was strongly correlated with SRF_PSMA_ (r = 0.872, *p* < 0.001) [28]. 

In addition, both ^99m^Tc-MAG3 and ^68^Ga-PSMA PET/CT studies identified morphological renal abnormalities such as hydronephrosis, obstruction and non-functional kidneys. The presence of these abnormalities was consistent between the two imaging techniques, with a correlation coefficient of 0.93. An example of kidney obstruction depicted on both ^99m^Tc-MAG3 scintigraphy and ^68^Ga-PSMA PET/CT is depicted in Figure 8, which shows poor visualisation of the non-functional right kidney on ^99m^Tc-MAG3 scintigraphy. This is confirmed on ^68^Ga-PSMA PET/CT images, which demonstrate significantly decreased tracer uptake and a dilated right-sided collecting system. ^68^Ga-PSMA PET/CT has the advantage of additional anatomical detail, and ^99m^Tc-MAG3 scintigraphy allows for the viewing of dynamic images that may detect additional pathologies such as reflux. 

On ^68^Ga-PSMA PET/CT, obstruction is depicted by stasis of the tracer in the collecting system (Figure 9), which corresponds to the rising renogram curve on ^99m^Tc-MAG3 renal scintigraphy. In addition, ^68^Ga-PSMA PET/CT confirms the presence of multiple renal cysts, which were detected as areas of relative photopenia in renal scintigraphy. 

This is similar to the findings of a recent study by Rosar et al., who depicted reno-ureteral obstruction by PSMA PET/CT imaging in most cases (92.9%) [29]. A study by Rassek et al. also demonstrated that relevant abnormalities of SRF_MAG3_ could be detected with sensitivities and specificities of 90% and 92% for SRF_PSMA_ [28]. 

The finding that an abnormal ^68^Ga-PSMA kidney morphology corresponds to abnormalities detected on ^99m^Tc-MAG3 renal scintigraphy has important clinical implications. This suggests that ^68^Ga-PSMA imaging could potentially serve as a valuable tool for assessing kidney morphology in these patients. Studies by Valind (18-F PSMA 1007) and Sarikanya et al. [31,32] comparing PSMA PET/CT and ^99m^TcDMSA showed a good correlation between the two studies and suggest that PET/CT can be used to detect localised functional defects such as scars. The higher spatial resolution of PET/CT allowed for the detection of small cysts that were not visualised on ^99m^Tc-DMSA SPECT and correctly identified false positive findings on SPECT due to cortical thinning [31], potentially offering a non-invasive alternative or complimentary investigation to ^99m^TcMAG3 renal scintigraphy. Several studies have shown that renal ^68^Ga-PSMA can reliably measure approximate renal parenchymal volume and cortical uptake correlates with renal function tests such as GFR and creatinine [33,34,35]. This is especially beneficial in cases where ^99m^Tc-MAG3 imaging may not be feasible, such as during Molybdenum-99 shortages [36], or when patient logistics preclude an additional study. It adds valuable additional information regarding kidney morphology since ^68^Ga-PSMA is routinely accompanied by CT, which allows for the detection of additional morphological abnormalities (such as renal cysts and dilated renal pelvis Figure 7) that are often not clearly depicted on renal scintigraphy. 

**Table 3 diagnostics-14-00578-t003:** Summarizes the reported correlations between different methods of split renal function assessment.

Study	Year	N	Method 1	Method 2	Correlation
Momin et al. [26]	2018	*n* = 50	^99m^Tc-DMSA	^99m^Tc-MAG3	0.99
Rassek et al. [28]	2023	*n* = 73	^18^F-PSMA	^99m^Tc-MAG3	0.872
Rosar et al. [29]	2020	*n* = 97	^68^Ga-PSMA	^99m^Tc-MAG3	0.91
Betz et al. [34]	2021	*n* = 50	^68^Ga-PSMA	^99m^Tc-DTPA	0.53
Present study	2023	*n* = 56	^68^Ga-PSMA	^99m^Tc-MAG3	0.957

Importantly, the split renal function on ^68^Ga-PSMA demonstrated high levels of agreement with ^99m^Tc-MAG3 split renal function in patients with both normal and impaired renal function (this remained true regardless of the PSA level)—see Figure 8, Figure 9 and Figure 10, which demonstrate results similar to the reports by Rosar et al. [29]. This study demonstrated that the correlation between PSMA- and MAG3-derived SRF was highly significant (*p* < 0.001), with a correlation coefficient of r = 0.91. 

Tumour sequestration affects ^68^Ga-PSMA accumulation in organs such as the kidneys. The tumour sink effect has, therefore, been suggested as a potential confounding factor in patients with high tumour volumes where kidney uptake appears decreased on ^68^Ga-PSMA PET/CT [29,37]. Conversely, Burgard et al. reported that there was no demonstrable tumour sink effect in the kidneys but confirmed this for the salivary glands and spleen [38]. In our study, the majority of patients had a high tumour burden but this did not limit our ability to accurately estimate DFR and detect morphological kidney abnormalities. However, no comparison was carried out post RLT to determine whether there was any change in renal function as this was beyond the scope of this study. It may be interesting to include this in future studies. 

Our findings have important practical and clinical implications for the management of patients with metastatic castration-resistant prostate cancer (mCRPC). ^68^Ga-PSMA PET/CT is an imaging modality that is commonly used to stage and monitor treatment response in this patient population. ^68^Ga-PSMA PET/CT also has the advantage of being performed after each cycle for monitoring treatment response; thus, kidney function can be simultaneously monitored in a serial fashion. Our study confirms the feasibility of using ^68^Ga-PSMA-PET/CT to calculate SRF at baseline for PSMA-based radioligand therapy (RLT) and to monitor interval changes. This approach may eliminate the need for additional ^99m^Tc-MAG3 scans, thereby reducing unnecessary radiation exposure to patients, additional hospital visits, and costs. 

## 5. Limitations

It is important to recognise the limitations of our study, which include its retrospective nature and relatively small sample size (*n* = 56). However, our findings are consistent with those reported on larger patient samples. A larger validation study is necessary to confirm our results and to further explore the utility of ^68^Ga-PSMA-derived split kidney function in clinical practice. 

## 6. Conclusions

^68^Ga-PSMA PET-derived split function demonstrated a high correlation with renal function assessed on diuretic ^99m^Tc-MAG3 renal scintigraphy. PET-derived split renal function may, therefore, be considered an alternative to diuretic renal scintigraphy-based split function. Furthermore, both ^99m^Tc-MAG3 renal scintigraphy and ^68^GaPSMA PET/CT studies identified morphological renal abnormalities such as hydronephrosis and shrunken kidneys. This correlation underscores the potential utility of ^68^Ga-PSMA imaging as a valuable tool for assessing kidney morphology as an alternative to renal scintigraphy-derived split renal function in clinical practice. 

## Figures and Tables

**Figure 1 diagnostics-14-00578-f001:**
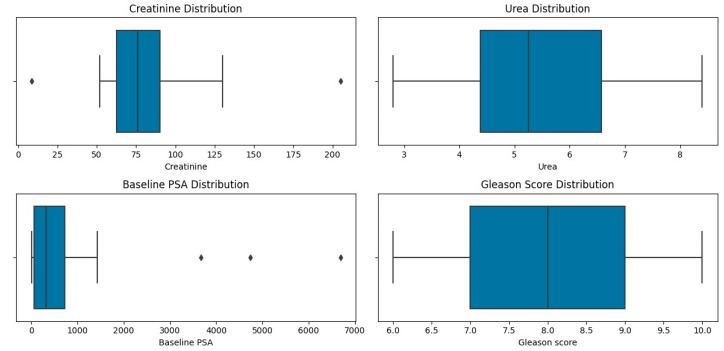
The distribution of creatinine, urea, PSA and Gleason score in the study population.

**Figure 2 diagnostics-14-00578-f002:**
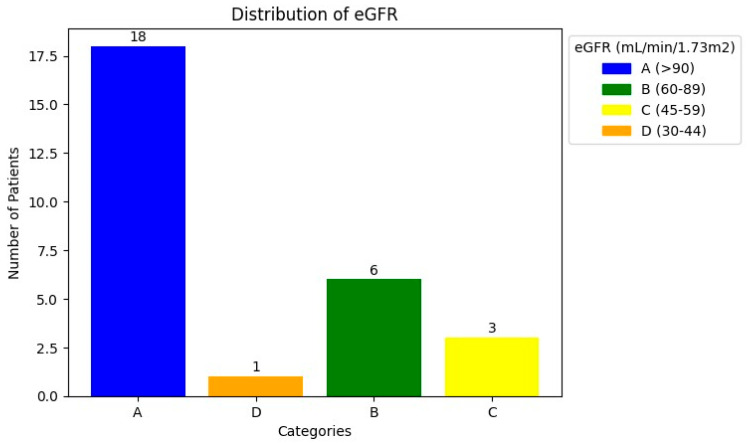
The distribution of eGFR in the study population.

**Figure 3 diagnostics-14-00578-f003:**
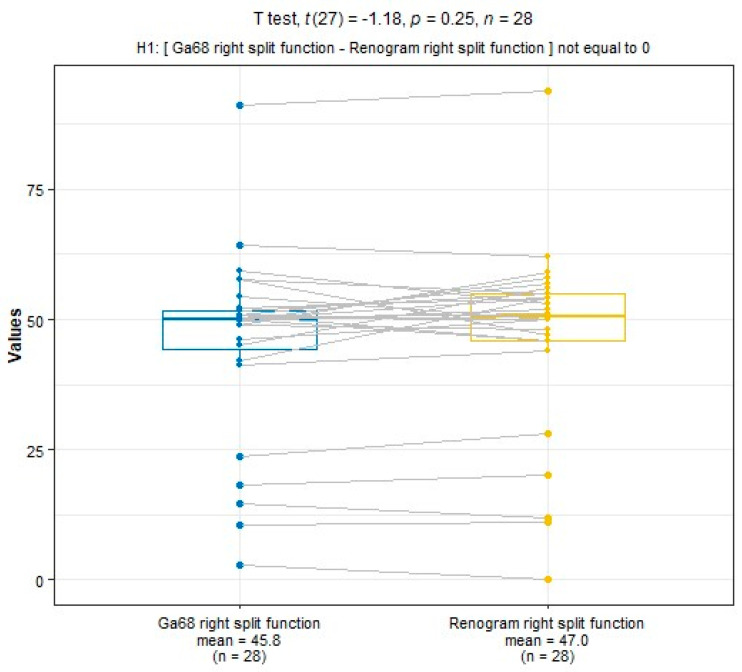
A positive correlation between the ^68^Ga-PSMA- and ^99m^Tc-MAG-3-derived SRF in the right kidney.

**Figure 4 diagnostics-14-00578-f004:**
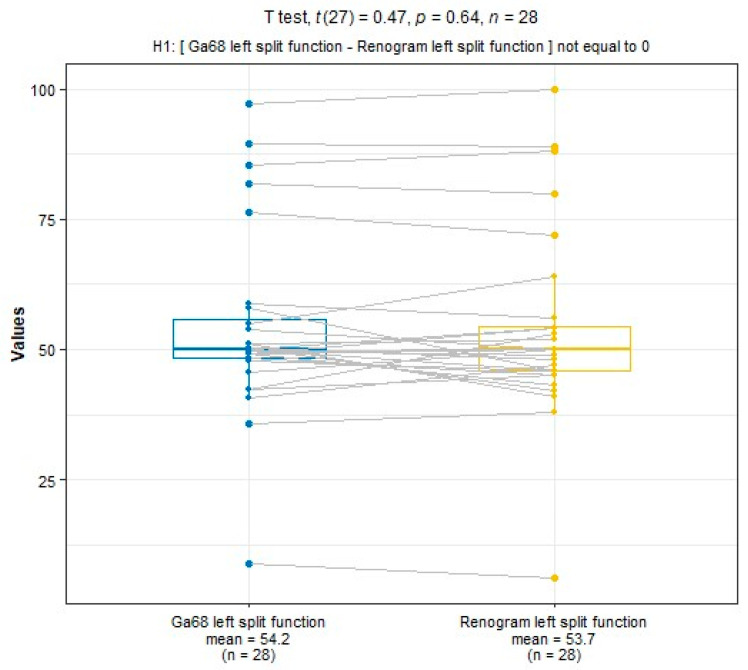
A positive correlation between the ^68^Ga PSMA- and ^99m^Tc-MAG-3-derived SRF in the left kidney.

**Figure 5 diagnostics-14-00578-f005:**
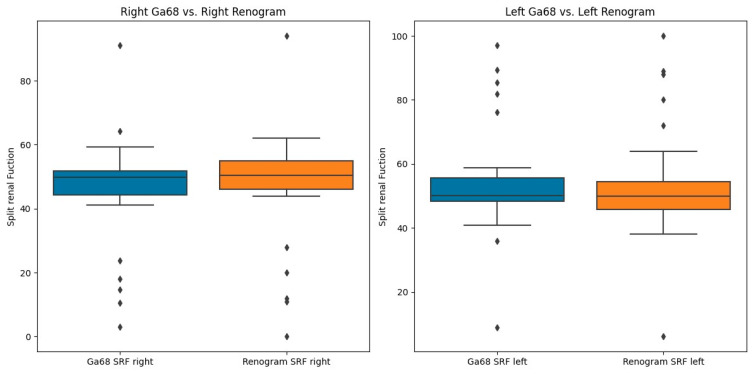
Boxplot showing the recorded differences between the renogram and the PET data. The SRF is centrally located with a few outliers.

**Figure 6 diagnostics-14-00578-f006:**
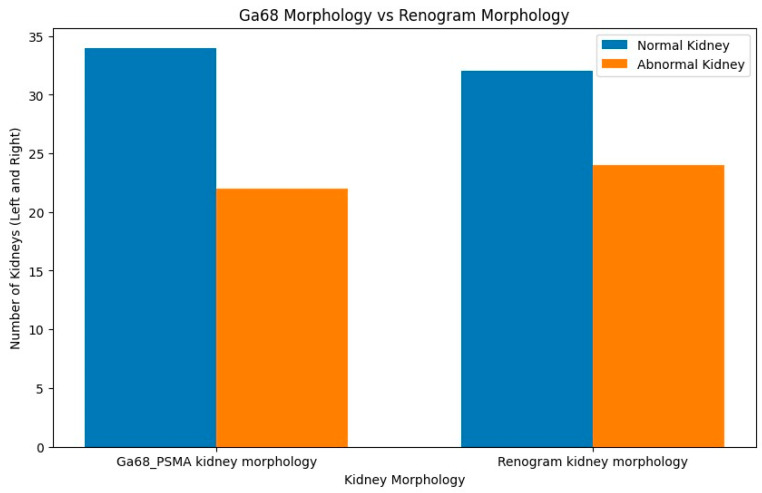
Histogram comparison for ^68^Ga-PSMA kidney morphology and ^99m^Tc-MAG3 kidney morphology, with a correlation coefficient of approximately 0.93.

**Figure 7 diagnostics-14-00578-f007:**
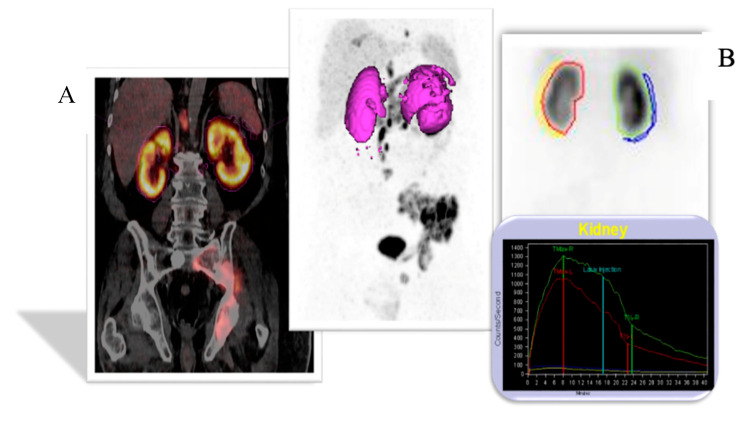
A case of a patient with normal renal function and equal SRF. (**A**) ^68^Ga PSMA PET; (**B**) ^99m^Tc-MAG3.

**Figure 8 diagnostics-14-00578-f008:**
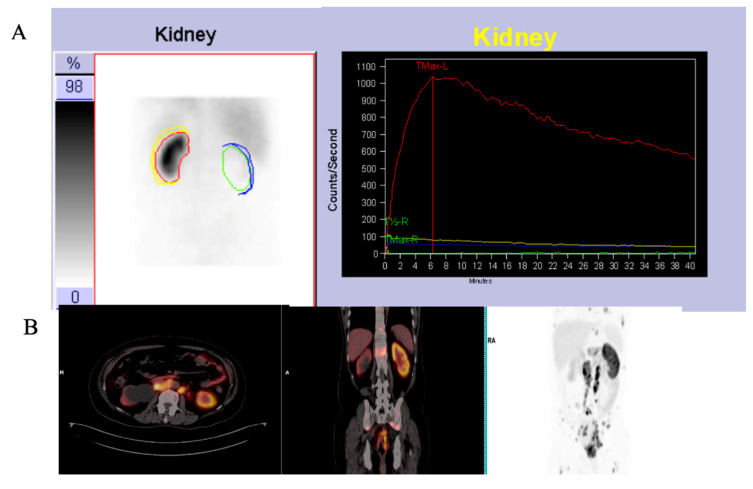
^99m^Tc-MAG3 renogram (**A**) showing an area of photopenia in a non-functioning right kidney, which is confirmed on the ^68^Ga-PSMA PET/CT images; (**B**) showing a right kidney with decreased tracer uptake and a dilated collecting system.

**Figure 9 diagnostics-14-00578-f009:**
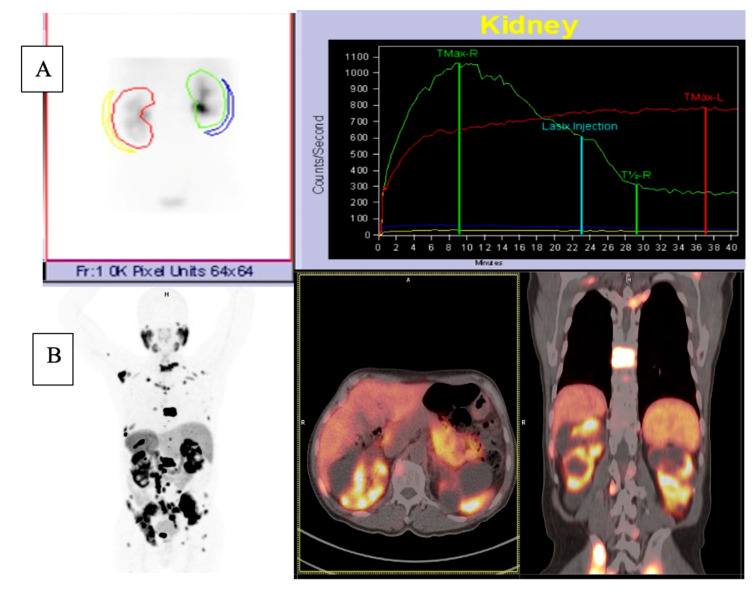
^99m^Tc-MAG3 renogram images (**A**) showing left kidney obstruction. ^68^Ga PSMA PET/CT images (**B**) showing multiple kidney cysts bilaterally with decreased tracer uptake and a stasis of radiotracer in a dilated collecting system.

**Figure 10 diagnostics-14-00578-f010:**
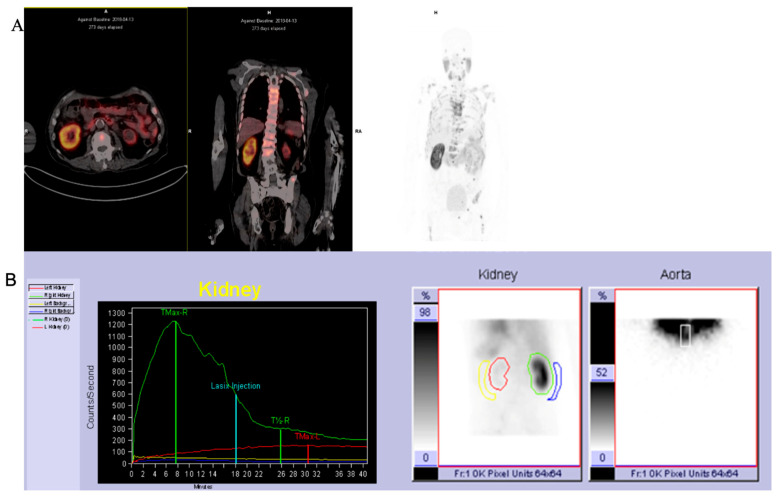
^68^Ga-PSMA PET/CT images (**A**) showing a shrunken left kidney with decreased tracer uptake. ^99m^Tc-MAG3 renogram (**B**) showing an area of photopenia in a non-functioning left kidney.

**Table 1 diagnostics-14-00578-t001:** Patient characteristics.

Number of patients	28
Age	(Years)
Min	45
Max	86
Mean	64
Median	67.5
Metastases	(*n*)
Bone	28
Lymph nodes	19
Lung	2
Others	1
eGFR	(mL/min)
<60 mL/min	*n* = 4 (CKD stage 3A *n* = 3; stage 3B *n* = 1)
≥60 mL/min	*n* = 24
Creatinine	µmol/L
Min	52
Max	205
Mean	81
Median	76
Urea	
Min	2.8
Max	8.4
Mean	5.20
Median	5.25
PSA	(ng/mL)
Min	3.95
Max	6685
Mean	869.81
Median	359.07

**Table 2 diagnostics-14-00578-t002:** Descriptive statistics.

	^68^Ga-PSMA PET/CT (Right)	99mTc- MAG3 (Right)	^68^Ga-PSMA PET/CT (Left)	99mTcMAG3 (Left)
Mean	45.8	47.0	54.2	53.7
Median	49.9	50.5	50	50
Std	17.8	18.4	17.8	18.4
Min	3	0.0	9	6
25%	44.4	46	48.3	45.8
50%	49.9	50.5	50.5	50
75%	51.7	55	55.6	54.5
Max	91	94	97	100

## Data Availability

Data are contained within the article.

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
