# Peer review of "A Comparison of 68Ga-PSMA PET/CT-Based Split Renal Function with 99mTc-MAG3 Renography in Patients with Metastatic Castration-Resistant Prostate Carcinoma Treated with 177Lu-PSMA"

_diagnostics, 2024, doi:10.3390/diagnostics14060578_

Round 1

Reviewer 1 Report

Comments and Suggestions for Authors

1.       There are some repetitions in the Introduction (e.g. on page 3: „Modern clinical management of prostate cancer relies on exploiting the prostate specific membrane antigen (PSMA) as a molecular target for both imaging and treatment of prostate cancer in patients who have failed other treatment modalities such as androgen deprivation therapy and chemotherapy after radical prostatectomy,” and on page 4: „Lu-177 PSMA RLT relies on exploiting the prostate specific membrane antygen (PSMA) as a molecular target for both imaging and treatment of prostate cancer in patients who have failed other treatment modalities such as androgen deprivation therapy and chemotherapy after radical prostatectomy.” Additionally, data on the effectiveness of 177Lu-PSMA are repeated) – it should be organized.

2.       Tab. 1: in 4 patients with eGFR<60 mL/min/1.73m2 at least stage of chronic kidney disease should be presented

3.       Fig. 1: eGFR distribution rather, instead of creatinine distribution should be showed

4.       Limitations of the study should be listed in the Discussion: at least retrospective character of the study, and quite low numer of cases included

5.       On Page 19 it is stated that the Institutional Review Board Statement was not applicable. It is at odds to the statement in Material & Methods section. That needs correction.

Author Response

Thank you for taking the time to review our work. This will help strengthen our study.

1.There are some repetitions in the Introduction (e.g. on page 3: „Modern clinical management of prostate cancer relies on exploiting the prostate specific membrane antigen (PSMA) as a molecular target for both imaging and treatment of prostate cancer in patients who have failed other treatment modalities such as androgen deprivation therapy and chemotherapy after radical prostatectomy,” and on page 4: „Lu-177 PSMA RLT relies on exploiting the prostate specific membrane antigen (PSMA) as a molecular target for both imaging and treatment of prostate cancer in patients who have failed other treatment modalities such as androgen deprivation therapy and chemotherapy after radical prostatectomy.” Additionally, data on the effectiveness of 177Lu-PSMA are repeated) – it should be organized.

 Response: Repetition on page 4 and on the effectiveness of 177Lu-PSMA               has been removed and data organized.

2. Tab. 1: in 4 patients with eGFR<60 mL/min/1.73m2 at least stage of  chronic  kidney disease (CKD) should be presented.

Response: Stages of CKD for the 4 patients added.

3. Fig. 1: eGFR distribution rather, instead of creatinine distribution should be showed. 

Response: A histogram showing eGFR distribution was added.

4.  Limitations of the study should be listed in the Discussion: at least retrospective character of the study, and quite low number of cases included.

Response: Limitations of the study were included in the discussions.

5.  On Page 19 it is stated that the Institutional Review Board Statement was not applicable. It is at odds to the statement in Material & Methods section. That needs correction.

Response: 'Institutional Review Board Statement was not applicable' -the statement was removed and replaced with "The study was conducted in accordance with Helsinki declaration and approved by the Human Research Ethics Committee of the University of KwaZulu Natal (protocol reference number: BREC /00003636/2021." 

Reviewer 2 Report

Comments and Suggestions for Authors

A comparison of 68Ga-PSMA PET/CT-based split renal function to 99mTc-MAG3 in patients with metastatic castration-resistant prostate carcinoma treated with 177Lu- PSMA

Introduction whicg is systematized provides sufficient information and also includes relevant refereneces. This study investigates the utility of PSMA-targeted imaging for determination of relative renal function and makes a comparison between  the split renal function using 68Ga-PSMA PET/CT and 99mTc-MAG-3 derived split renal function which was done in 56 kidneys. Conclusion are supported by the result so that PSMA derived split function demonstrated a high correlation with renal function assessed on diuretic 99mTc-MAG3 renograms. Methods are adequately described and include a retrospective cross-sectional study involving 28 patients with metastatic castration-resistant prostate cancer (mCRPC) who previously received 1 st and 2nd line chemotherapy in addition to Androgen deprivation therapy, who underwent 177LuPSMA-617 radioligand therapy between June 2019 – September 2023. Histograms, doagrams and boxplotes are used to present recorded differenecs between renograms. Therefore study design is appropriated.

Author Response

Thank you very much for taking the time to review our work.